# Mimic Pork Rinds from Plant-Based Gel: The Influence of Sweet Potato Starch and Konjac Glucomannan

**DOI:** 10.3390/molecules27103103

**Published:** 2022-05-12

**Authors:** Qibo Zhang, Lu Huang, He Li, Di Zhao, Jinnuo Cao, Yao Song, Xinqi Liu

**Affiliations:** 1National Soybean Processing Industry Technology Innovation Center, Beijing Technology and Business University (BTBU), Beijing 100048, China; zhangqibo2021@163.com (Q.Z.); huanglulu1119@163.com (L.H.); zhaodi22121@163.com (D.Z.); 2Plant Meat (Hangzhou) Health Technology Limited Company, Hangzhou 311121, China; jinnuocao@163.com; 3Handan Institute of Innovation, Peking University, Handan 056008, China; 2005songyao@163.com

**Keywords:** plant-based pork rinds, composite gel, texture, rheology, gelling mechanism

## Abstract

This study investigated the effect of sweet potato starch (SPS) and konjac glucomannan (KGM) on the textural, color, sensory, rheological properties, and microstructures of plant-based pork rinds. Plant-based gels were prepared using mixtures of soy protein isolate (SPI), soy oil, and NaHCO_3_ supplemented with different SPS and KGM concentrations. The texture profile analysis (TPA) results indicated that the hardness, cohesiveness, and chewiness of the samples improved significantly after appropriate SPS and KGM addition. The results obtained via a colorimeter showed no significant differences were found in lightness (L*) between the samples and natural pork rinds after adjusting the SPS and KGM concentrations. Furthermore, the rheological results showed that adding SPS and KGM increased both the storage modulus (G’) and loss modulus (G’’), indicating a firmer gel structure. The images obtained via scanning electron microscopy (SEM) showed that the SPS and KGM contributed to the formation of a more compact gel structure. A mathematical model allowed for a more objective sensory evaluation, with the 40% SPS samples and the 0.4% KGM samples being considered the most similar to natural pork rinds, which provided a comparable texture, appearance, and mouthfeel. This study proposed a possible schematic model for the gelling mechanism of plant-based pork rinds: the three-dimensional network structures of the samples may result from the interaction between SPS, SPI, and soybean oil, while the addition of KGM and NaHCO_3_ enabled a more stable gel structure.

## 1. Introduction

Pork rinds are popular snacks in many countries and regions, including the USA, Australia, Europe, and Asia [1,2]. As the plant-based market has prospered in recent years, plant-based products have been further subdivided to meet different consumer demands, expanding into the pork rind field. Compared to traditional pork rinds, plant-based pork rinds are environmentally friendly, avoiding the environmental pollution and greenhouse gas emissions produced from farming. The preparation of plant-based pig skins does not the slaughtering of pigs, which improves animal welfare [3]. Furthermore, meat production requires an additional energy loss level, while developing plant-based pork rinds without animal ingredients is more resource-efficient and sustainable [4]. Since pork rinds are produced from the part of the pig directly in contact with external pathogens, they are likely to cause foodborne diseases if they are not thoroughly cleaned during processing [5]. The residual grease from pork rinds is challenging to clean, and the traditional manufacturing process necessitates repeated washing with chemicals, which undoubtedly creates a safety hazard for consumers [6]. Therefore, plant-based pork rinds are gaining popularity due to their benefits for the planet and human well-being.

However, the plant-based pork rinds currently available on the market are primarily available in one form, puffed pork rinds. Outstanding Foods uses rice, sunflower oil, and pea protein as ingredients to create plant-based puffed pork rinds via grilling (announced on its official website). Snacklins has launched a plant-based crisp that uses cassava to achieve a puffed pork rind-like texture and creates a meaty flavor using mushrooms (announced on its official website). Minimal studies are available involving edible plant-based pork rinds that accurately mimic the original texture, appearance, and mouthfeel of natural pork rinds. There is a significant overlap between food science and medicine as a multidisciplinary field [7]. In the medical field, gels are commonly used to simulate skin for wound management, cosmetic surgery [8], and in vitro simulation experiments [9]. Natural gels are non-toxic, biocompatible, and biodegradable and can be obtained from various sources [10]. The current commercial skin substitutes, Integra and Matriderm, use natural hydrogels as biological scaffolds that not only provide a friendly environment for cells and load biomolecules but also retain water to promote cell migration and proliferation [11]. Zhang et al. mimicked the viscoelastic properties of porcine skin using agarose gels [12]. Although skin simulation gels are primarily used for medical purposes at this stage, they show significant potential for extrapolation to the food field.

By utilizing the synergy between the ingredients, composite gels can compensate for the limitations of single-ingredient gels to better imitate pork rinds [13,14,15]. Soybeans represent a high-quality plant source that contains proteins and lipids with high nutritional benefits [16], as well as excellent functional properties like emulsification and gelation [17]. Sweet potatoes are rich in dietary fiber containing more than ten micronutrients and little cholesterol or lipids and are considered an ideal natural food by nutritionists [18]. Sweet potato starch (SPS) is the main product of sweet potatoes and forms a complete three-dimensional gel structure to ensure its firmness [19]. Konjac glucomannan (KGM) is a natural polymer polysaccharide extracted from konjac, which is popular for its positive health effects, such as anti-obesity, antioxidant, and hypoglycemic activity [20]. KGM is widely used in the food industry due to its excellent biocompatibility, thickening, and gelling properties [21]. Studies have shown that KGM displays good synergy with protein and starch [22,23], exhibiting potential for further developing composite gel products.

This research aims to prepare plant-based gels that can better mimic the texture, appearance, and mouthfeel of natural pork rinds and optimize these properties by adjusting the SPS and KGM concentrations. Furthermore, the interaction mechanisms and microstructures of the plant-based gels were examined via rheometry and scanning electron microscopy (SEM).

## 2. Results and Discussion

### 2.1. Textural Analysis

The textural properties were crucial for measuring the similarity of the samples to natural pork rinds. The textural parameters (hardness, cohesiveness, springiness, and chewiness) of the plant-based pork rinds containing different SPS and KGM concentrations are presented in Table 1, except for the10% SPS samples which were too soft to form the specified shape.

The hardness, cohesiveness, and chewiness of the samples in the SPS group increased at higher concentrations while those in the KGM group were initially higher, followed by a decline as the concentrations increased. Studies have shown that amylose concentration is closely related to gel strength [24]. Therefore, the addition of SPS resulted in higher amylose content and firmer gels. Schwartz et al. reported that low KGM concentrations affected gelatinization and starch retrogradation, but this effect is barely noticeable at high KGM concentrations [25], which is consistent with the experimental results. The 0.2% KGM samples were softer than those without KGM, possibly because KGM forms a complex with leached amylose that cannot fill the gel network structure well [26]. However, increasing the KGM concentration from 0.2% to 0.8% increases the number and volume of complexes, which are more successful in filling and strengthening the gel structure [27].

All the samples could better simulate the springiness of natural pork rinds boiled for 30 min. In the SPS group, no significant differences were evident between the chewiness of the samples with 40% SPS and natural pork rinds. Although the hardness of the samples in the KGM group was exceedingly similar to natural pork rinds, the chewiness was significantly different from natural skin even when the KGM concentration was increased to 0.8%. This may be because the KGM group was based on 30% SPS, indicating that the SPS concentrations more significantly affected the textural properties of plant-based pork rinds than KGM.

### 2.2. Color Analysis

Color plays a crucial role in food acceptability and consumer preference [28]. Determining the lightness (L*), redness/greenness (a*), and yellowness/blueness (b*) color parameters of the samples reflected the similarities in appearance between the plant-based pork rinds and natural pork rinds. The 10% SPS samples that were too soft to form the specified shape were not involved in the determination.

As shown in Figure 1, the addition of starch significantly changed the color parameters, especially L*. Although the L* value decreased as the SPS concentration increased from 20% to 40%, it dramatically increased when the added SPS exceeded 40% (Figure 1A). This may be because the starch is too dry to sufficiently gelatinize with a low moisture content [29], resulting in white lumps in some parts of the samples. Contrary to this, as shown in Figure 1B, KGM abruptly increased the L* value, followed by a gradual decrease, while exhibiting a negligible effect on a* and b* values. The color of the plant-based pork rinds was generally more similar to natural pork rinds boiled for 30 min than raw pork rinds.

### 2.3. Sensory Evaluation

The fuzzy mathematics sensory evaluation can objectively and precisely distinguish the merit levels of different products and select the best product from several samples according to the grade membership degree and membership function theory [30].

Based on the preliminary sensory results (Table 2), the fuzzy matrix R was formed via the fuzzy mathematical method. Matrix multiplication was used to calculate the evaluation matrix *Y*, while the degrees of membership for each grade in *Y* were assigned and accumulated to obtain the sensory comprehensive score (SCS) of the samples (Figure 2A).

The SCS of the SPS and KGM groups increased initially, followed by a decrease. As shown in Figure 2B, the 10% SPS gels were soft and sticky with a pasty texture, which was difficult to associate with natural pork rinds. The SCS of the samples increased at higher SPS concentrations, peaking at 40% SPS due to improved appearance and texture. Without KGM, the samples were difficult to shape during steaming, while a water film covered the surface, resulting in high viscosity. The addition of KGM reduced the stickiness of the samples, improved the gel strength, and caused the formation of KGM-SPS complexes. As the number of complexes increased, the appearance of plant-based pork rinds became rougher (Figure 2C), presenting a distinct grainy mouthfeel, especially in the 0.6–0.8% KGM samples, which exhibited decreased SCS. Therefore, the 40% SPS and the 0.4% KGM samples were considered the closest to the natural pork rinds in texture, appearance, and mouthfeel.

### 2.4. Rheological Analysis

#### 2.4.1. Frequency Scanning

As shown in Figure 3, the samples displayed higher storage modulus (G’) and loss modulus (G’’) as the frequency increased, while each curve showed varying degrees of frequency dependence. The addition of SPS significantly increased the two moduli (Figure 3A,B). This may be because a higher SPS concentration promotes the interaction and cross-linking between starch molecules, making the network structure more compact [31]. The G’ of the 50% SPS samples was lower than the 40% SPS samples, while the G’’ of the former was higher than the latter. Therefore, it is assumed that the low moisture in the 50% SPS samples inhibits starch gelatinization, resulting in less elastic components and more viscous components in the gel system [29,32]. Figure 3C,D showed that the G’ and G’’ increased with the increasing KGM concentration, indicating that KGM contributed to a stronger network structure. Similar results were observed by Ning et al. [26]. These findings were also consistent with those of Ma et al., who revealed that KGM molecules combined with leached amylose to form a stronger gel structure [33].

#### 2.4.2. Temperature Scanning

Similar to the frequency scanning results, the G’ and G’’ increased at higher SPS and KGM concentrations (Figure 4), suggesting a promotional effect on the gel structure. The G’ and G’’ generally showed a downward trend throughout the heating process. At 25–75 °C, the two moduli of the samples decreased gradually. At 75 °C, the moduli of the 30% SPS samples and the high KGM concentration groups (0.4%, 0.6%, and 0.8%) showed a brief rise, which could be attributed to the interaction between SPS, KGM, and NaHCO_3_. In a heated alkaline environment, the KGM molecules were prone to deacetylation, forming more hydrogen bonds between the deacetylated KGM and SPS, temporarily facilitating a firmer gel structure [21]. Similarly, Luo et al. showed that the interaction between the alkali and KGM rapidly increased the two moduli over 70 °C [34]. However, the hydrogen bonds formed at high temperatures were unstable. Further heating ruptured the hydrogen bonds between the molecules, collapsing the structures and significantly decreasing G’ and G’’ [35].

### 2.5. Microstructure

The three-dimensional network structure of the samples containing different SPS and KGM concentrations was observed via SEM (Figure 5A–D). The SPS group exhibited smaller holes and denser structures due to a higher SPS concentration, forming a firmer gel, which was consistent with the TPA results (Table 1). In addition, granules were evident on the surfaces of the 50% SPS samples (Figure 5B), which were likely insufficiently gelatinized SPS particles. A comparison between Figure 5C,D showed that the addition of KGM filled the pores with KGM-SPS complexes, improving the hardness and cohesiveness of the samples while producing a chewier gel [36].

The microstructures of the natural pork rinds are shown in Figure 5E,F. The grease in the pores of the raw pork rinds was removed by hydrothermal treatment for 30 min to obtain a clear collagen pore structure. Since the plant-based pork rinds showed comparable structures to natural pork rinds, with similar pore sizes (10–60 μm) and depths, their texture and mouthfeel were similar.

### 2.6. Schematic Model

Based on the experiments and references mentioned above, this study proposed a possible schematic model for the gelling mechanism of plant-based pork rinds (Figure 6A). The primary components in the emulsion were soy protein isolate (SPI), soybean oil, SPS, NaHCO_3_, and KGM. During heating, the SPS molecules absorbed water, causing them to swell and even disintegrate to release amylose. The amylose cross-links formed three-dimensional network structures during cooling [19,29]. Likewise, SPI molecules were denatured by heating, forming cross-linked aggregates to produce the network structure [37]. Previous research showed that soybean oil combined with the hydrophobic groups in SPI, promoting the aggregation of oil and protein [28]. Moreover, the alkaline environment created by the NaHCO_3_ increased the size of starch granules and made them more prone to breakage [38,39]. However, the swelling of the starch granules was restricted in the presence of KGM [39]. On the one hand, KGM competed with starch for water molecules and reduced the growth of the starch granules [40]. On the other hand, the acetyl groups were removed from KGM in a heated alkaline environment [21,41]. The deacetylated KGM molecules displayed stronger interaction with amylose to form KGM-SPS complexes, which were embedded into the network structure during cooling. To sum up, the three-dimensional network structure of the plant-based pork rinds may result from the interaction between SPS, SPI, and soybean oil, while the addition of KGM and NaHCO_3_ increases the stability of the gel structure.

As shown in Figure 6B, the internal layout of the plant-based pork rinds can be further inferred according to the proposed gelling mechanism. The SPS occupies most of the space, followed by the SPI combined with soybean oil. The KGM that surrounded the granules could bind to the amylose released from starch, while NaHCO_3_ could create an alkaline environment to promote the deacetylation of KGM. Changes in the SPS and KGM concentrations affect the internal layout of the samples, representing a crucial element of their macroscopic characteristics.

In the 10% SPS samples, the low starch concentration created significant distances between the retrograded starch granules and large gaps in the gel structure. Although the SPI network structure had enough space to stretch, the low content prevented sufficient gel strength. The excess KGM molecules could not bind to the SPS, consequently entering a free state. Therefore, the 10% SPS samples were paste-like and displayed the worst textural properties. Contrarily, the 50% SPS samples exhibited the most crowded interiors, to the extent that the SPS compressed the SPI network structure space. The high starch concentration significantly enhanced the gel structure, as shown by the TPA results (Table 1). The alkaline environment created by the NaHCO_3_ in the absence of the synergistic KGM effect promoted SPS granule swelling and rupture in the 0% KGM sample, compromising the gel strength. The sample is also highly sticky due to the transudatory amylopectin. A high KGM concentration caused the formation of complexes that filled the pores, strengthened the structure, reduced stickiness, and even allowed the KGM to enter a free state. The remaining acetyl groups in the KGM increased due to the relatively limited alkaline environment produced by NaHCO_3_.

## 3. Materials and Methods

### 3.1. Materials

The SPI (protein = 91.2%, moisture = 5.1%, on a dry basis) was purchased from the Shandong Yuwang Ecological Food Industry Co., Ltd. (Shandong, China), while the soy oil was obtained from the China Oil & Foodstuffs Corporation Co., Ltd. (Beijing, China). The SPS (starch = 90.2%, moisture = 8.6%, on a dry basis) and edible NaHCO_3_ were purchased from the Beijing Shuntian Heng Trading Co., Ltd. (Beijing, China). The konjac powder (glucomannan = 83.6%, moisture = 10.5%, and ash = 3.8%, on a dry basis) was acquired from the Hubei Consistent Biotechnology Co., Ltd. (Hubei, China). Fresh pork rinds (abdomen) were obtained from a local market (Beijing, China). All of the external fat was removed from the pork rinds, which were stored at 4 °C in a fridge before testing.

### 3.2. The Preparation of the Plant-Based Pork Rind Emulsion

The SPI powder (1%, *w*/*w*) was diluted with distilled water and stirred with a magnetic stirrer until evenly dispersed. The SPI solution was then decanted into an electronic stirrer (EUROSTAR 40 digital, IKA, Baden-Württemberg, Germany) to continue stirring for 5 min at 8000 rpm. The soy oil (1.5%, *w*/*w*) was added and stirred for 5 min at 8000 rpm to emulsify. The SPS powder and NaHCO_3_ (0.15%, *w*/*w*) were dissolved in distilled water, poured into the electronic stirrer, and blended at 6000 rpm for 5 min. The konjac flour was gradually added and stirred at 3000 rpm for 5 min to obtain the final plant-based pork rind emulsion. Two sample groups were used in this study, namely the SPS group: 0.4% KGM and different SPS concentrations (10%, 20%, 30%, 40%, and 50%, *w*/*w*) and the KGM group: 30% SPS and different KGM concentrations (0%, 0.2%, 0.4%, 0.6%, and 0.8%, *w*/*w*).

### 3.3. The Preparation of the Plant-Based Pork Rind Gel

To shape the emulsion, it was decanted into a specific baking mold (a cube, width 3 cm, length 10 cm, and height 4 mm) and steamed for 5 min, ensuring an average thickness of about 4 mm. Then, the preliminary gel in the mold was packed into a vacuum bag (width 8.9 cm and length 14.7 cm) and degassed at −0.09 MPa for 15 s using a vacuum machine (Exelway, DZ-300, Quanzhou Liding Mechanical Equipment Co., Ltd., Fujian, China), boiled at 95 °C for 15 min in a thermostat water bath (Jintan Kexi, HH-2 Water bath pot, Jintan Kexi Instrument Co., Ltd., Jiangsu, China), and finally retrograded in a 40 °C drying oven for 12 h to form a stable gel structure.

### 3.4. TPA

The textural properties of the samples were determined according to a method described by Xu et al. with some modifications [42]. A texture analyzer (CT3, Brookfield, Middleboro, MA, USA) equipped with a TA10 probe (diameter 12.7 mm, length 35 mm, and weight 5 g) was used for TPA. All of the samples were cut into 30 mm × 30 mm × 4 mm cubes, after which only the centers were compressed using the probe. The test parameters were set as follows: a reduced distance of 3 mm, a trigger point load of 5 g, and pre-test, test, and post-test speeds of 1.0 mm/s. The textural parameters, including hardness, cohesiveness, springiness, and chewiness, were obtained via the instrument software TexturePro CT.

### 3.5. Color Measurements

A colorimeter (CM-3610, Konica Minolta, Japan) was used to determine L*, a*, and b* color parameters of the samples. Before testing, the instrument was calibrated with a whiteboard and a standard blackboard (provided by Konica Minolta), and all samples were cut into 30 mm × 30 mm × 4 mm cubes. Five measurements were performed for each sample, and the L*, a*, and b* values were recorded.

### 3.6. Preliminary Sensory Evaluation

The sensory panelists consisted of ten participants with food knowledge and tasting experience. The sensory panelists were asked to rate the appearance, chewiness, stickiness, and cohesiveness of the plant-based rinds according to four grades (very good, good, medium, and bad). The samples were equilibrated at room temperature (26 °C) for 1 h before the sensory evaluation. After each sample evaluation, the panelists were requested to rinse their mouths with distilled water to avoid the experimental errors caused by the residual taste of the previous sample.

### 3.7. Fuzzy Mathematics Sensory Evaluation

A fuzzy mathematics sensory evaluation model was established according to a procedure previously described by Xue et al. to quantify the evaluation [43]. 

First, according to the quality attributes and scoring grades, the factor set was expressed as *U* = {u_1_, u_2_, u_3_, u_4_}, in which u_1_, u_2_, u_3_, and u_4_ represented appearance, chewiness, stickiness, and cohesiveness, while the grade set *V* = {v_1_, v_2_, v_3_, v_4_}, in which v_1_, v_2_, v_3_, and v_4_ represented very good (10 points), good (7 points), medium (4 points), and bad (1 point).

Second, the preliminary sensory evaluation results were transformed into a fuzzy matrix *R* using the fuzzy mathematical method.

Moreover, to determine the weight set *A*, 15 assessors (five professionals in the field of plant-based meat and ten researchers with food knowledge) were asked to make one-to-one comparisons between the importance of each quality attribute. In the three categories denoting importance, “0:4”, “1:3”, and “2:2” represented “a large difference”, “a little difference”, and “no difference”, respectively. The scores of each quality attribute were accumulated and normalized to obtain weight set *A* = {0.235, 0.338, 0.222, 0.205}, where 0.235, 0.338, 0.222, and 0.205 correspond to the weight of appearance, chewiness, stickiness, and cohesiveness, respectively.

Finally, the overall sensory comprehensive score (SCS) was calculated using Equations (1) and (2):*Y*_j_ = *A* × *R*_j_ = [y_1_, y_2_, y_3_, y_4_](1)
SCS_j_ = 10 × y_1_ + 7 × y_2_ + 4 × y_3_ + 1 × y_1_(2)

*Y* was the evaluation matrix, in which y_1_, y_2_, y_3_, and y_4_ represented the membership degrees for the grades of “very good”, “good”, “medium”, and “bad”, respectively, while j represented the jth sample.

### 3.8. Rheometry

The rheological experiments were performed according to a method delineated by Huang et al. with minor modifications [44]. The emulsion of the plant-based pork rind was heated at 95 °C and stirred at 4000 rpm for 5 min in a cooking machine (Vorwerk, TM5, Vorwerk, Wuppertal, Germany) and cooled to room temperature (25 °C) before measurement. A stress-controlled rheometer (DHR-1, TA Instruments, New Castle, DE, USA) equipped with a Peltier temperature control device was used to determine the rheological properties of the samples. A sample of approximately 4 g was placed in the center of the parallel geometric plate (diameter 40 mm), and the gap between the two plates was set to 1 mm. The specimen excess was removed from the plates using a scraper, while a thin layer of methyl silicone oil was used to prevent evaporation from exposed free edges of the sample. The sample was equilibrated for 5 min before each measurement.

Frequency scanning (strain 1%, temperature 37 °C) was performed to determine the variation in G’ and G’’ at different frequencies from 1.0 Hz to 10.0 Hz. Temperature scanning (angular frequency 10 rad/s, strain 1%) was performed to determine the variation in the G’ and G’’ at different temperatures from 25 °C to 95 °C.

### 3.9. SEM

The samples were cut into 4 mm cubes, pre-frozen in a refrigerator at −20 °C for 48 h, and dried in a vacuum freeze dryer (Marin Christ, Beta 1–8 LSC basic, Christ, Osterode, Germany) for 48 h. The prepared samples were then attached to a metal holder and coated with gold. The SEM images were obtained using a scanning electron microscope (Zeiss Gemini 300 SEM, Carl Zeiss, Jena, Germany) at an accelerating voltage of 1.5 kV.

### 3.10. Statistical Analysis

The statistical analysis was performed using SPSS 25.0, while the graphs were prepared using GraphPad Prism 9.0.0. An analysis of variance (ANOVA) was used to determine significant differences between the results at a significance level of 0.05.

## 4. Conclusions

Based on the above results, we conclude that a plant-based gel can mimic natural pork rinds while adjusting the SPS and KGM concentrations optimizes the texture, appearance, and mouthfeel of plant-based pork rinds. The gelling mechanism of the plant-based pork rinds is hypothesized: the three-dimensional network structures of the samples result from the interaction between the SPS, SPI, and soybean oil with the KGM-SPS complexes filling the pores and the NaHCO_3_ promoting KGM and SPS interaction. Future research intends to validate this hypothesis by further investigating the effect of SPI, soy oil, and NaHCO_3_ concentrations on plant-based pork rinds. Moreover, the insight into the quantitative sensory parameters of plant-based pork rinds can be increased by introducing oral tribology.

## Figures and Tables

**Figure 1 molecules-27-03103-f001:**
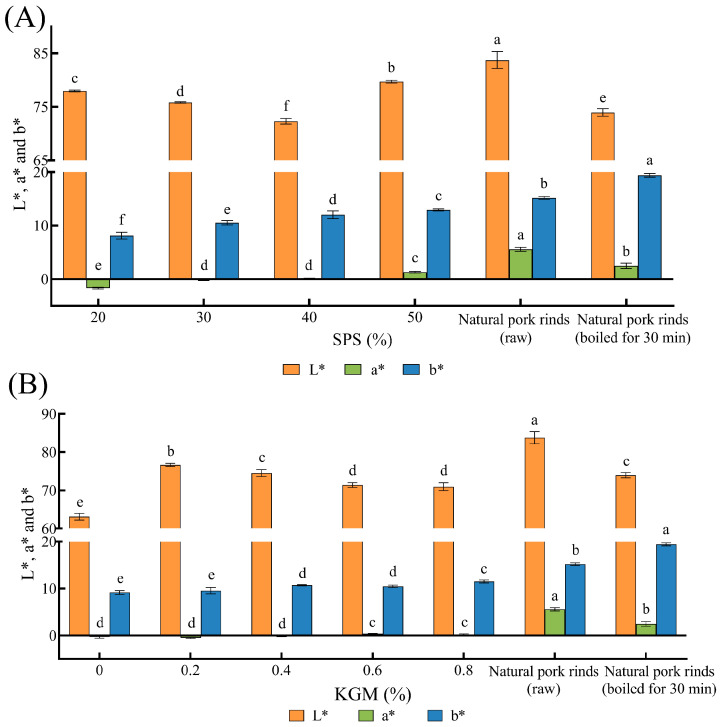
The color parameters of the plant-based pork rinds containing different SPS (**A**) and KGM (**B**) concentrations. Different lowercase letters (a–f) in the same color group denote significant differences (*p* < 0.05).

**Figure 2 molecules-27-03103-f002:**
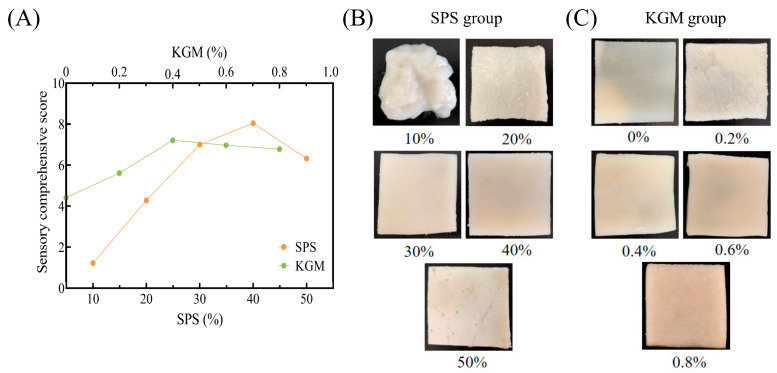
The SCS (**A**) and photographic images (**B**,**C**) of the plant-based pork rinds containing different SPS and KGM concentrations.

**Figure 3 molecules-27-03103-f003:**
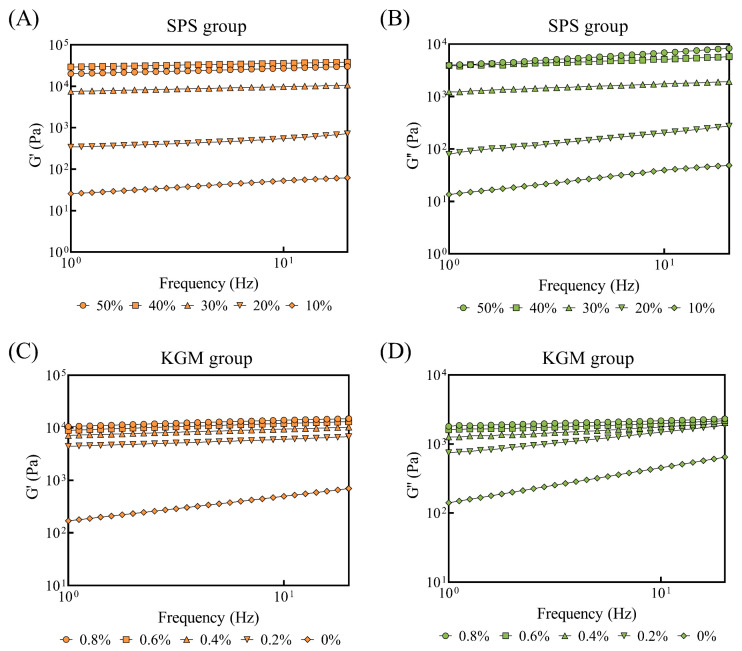
The G’ and G’’ of the plant-based pork rinds (frequency scanning) containing different SPS (**A**,**B**) and KGM (**C**,**D**) concentrations.

**Figure 4 molecules-27-03103-f004:**
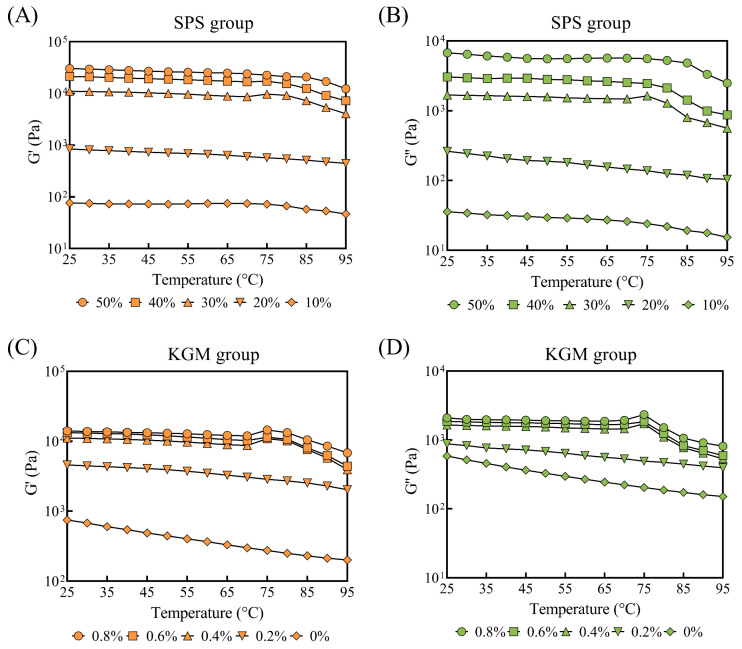
The G’ and G’’ of the plant-based pork rinds (temperature scanning) containing different SPS (**A**,**B**) and KGM (**C**,**D**) concentrations.

**Figure 5 molecules-27-03103-f005:**
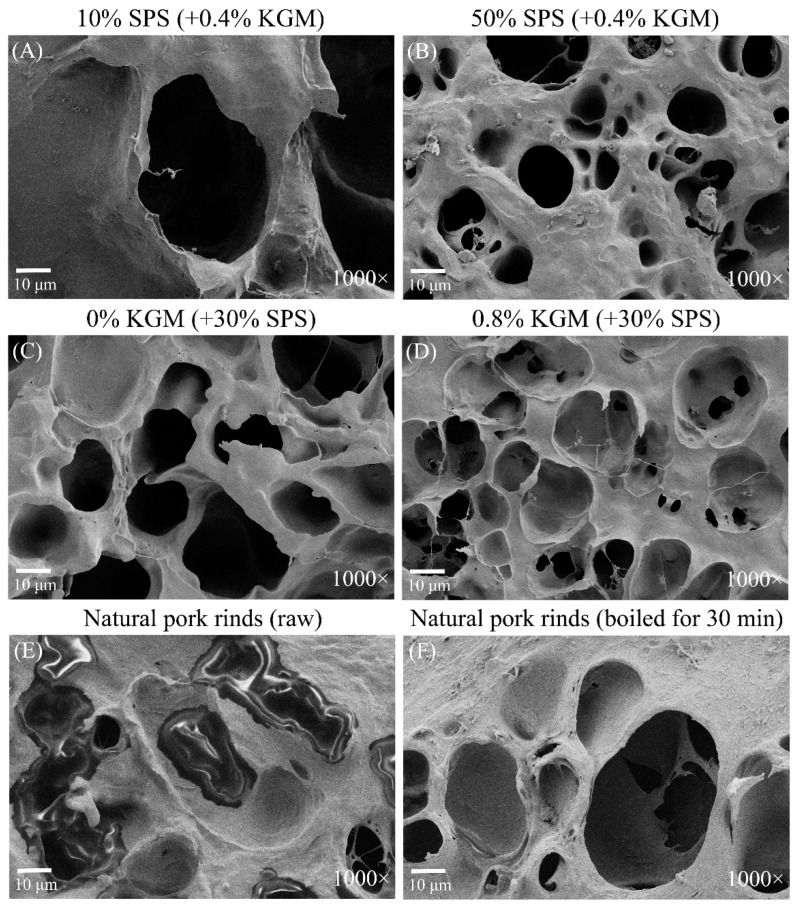
The microstructures of the plant-based pork rinds containing different SPS (**A**,**B**) and KGM (**C**,**D**) concentrations and natural pork rinds (**E**,**F**).

**Figure 6 molecules-27-03103-f006:**
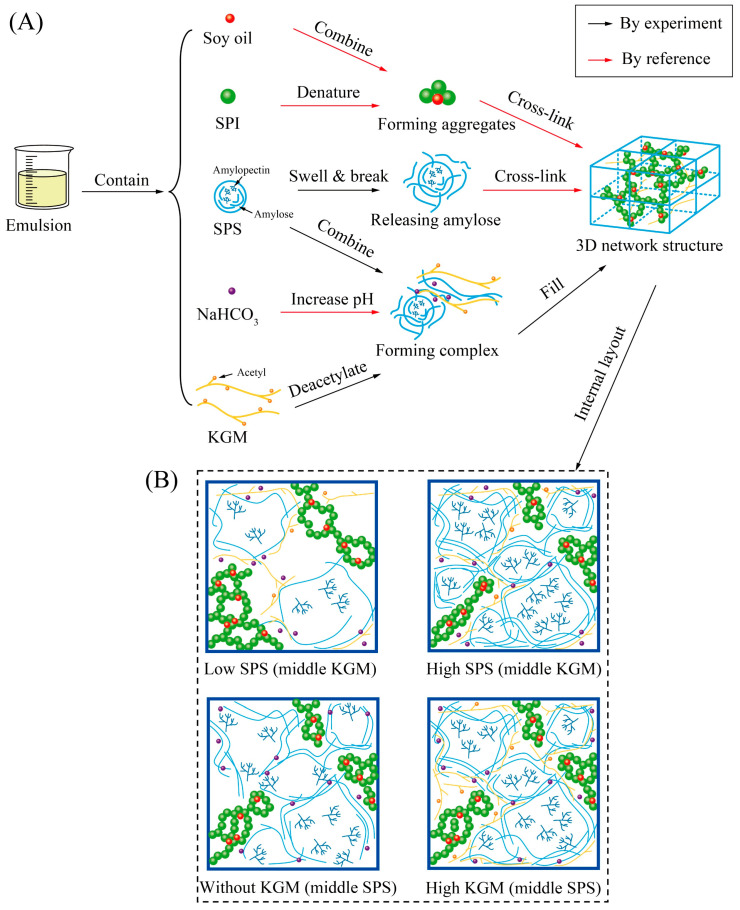
A schematic model showing the gelling mechanism (**A**) and internal layout (**B**) of the plant-based pork rinds containing different SPS and KGM concentrations.

**Table 1 molecules-27-03103-t001:** The Texture profile analysis (TPA) of the plant-based pork rinds containing different SPS and KGM concentrations.

Type and Proportion	Hardness (g)	Cohesiveness	Springiness	Chewiness (mJ)
SPS				
10%	none	none	none	none
20%	312.67 ± 28.57 ^d^	0.66 ± 0.03 ^c^	0.93 ± 0.01 ^a^	5.67 ± 0.71 ^d^
30%	1941.67 ± 42.72 ^c^	0.55 ± 0.02 ^d^	0.91 ± 0.01 ^a^	28.63 ± 0.38 ^c^
40%	3098.67 ± 60.19 ^b^	0.64 ± 0.03 ^c^	0.92 ± 0.03 ^a^	53.59 ± 0.61 ^b^
50%	6681.33 ± 122.27 ^a^	0.76 ± 0.02 ^b^	0.93 ± 0.02 ^a^	138.67 ± 2.38 ^a^
Natural pork rinds (boiled for 30 min)	2078.67 ± 55.81 ^c^	0.93 ± 0.02 ^a^	0.92 ± 0.03 ^a^	52.27 ± 1.64 ^b^
KGM				
0%	2181.67 ± 122.52 ^b^	0.60 ± 0.03 ^c^	0.89 ± 0.02 ^a^	34.82 ± 1.23 ^c^
0.2%	1941.67 ± 42.72 ^c^	0.52 ± 0.02 ^d^	0.91 ± 0.01 ^a^	26.70 ± 0.46 ^e^
0.4%	2134.33 ± 143.40 ^b^	0.54 ± 0.01 ^d^	0.91 ± 0.04 ^a^	30.04 ± 1.44 ^d^
0.6%	2138.33 ± 25.38 ^b^	0.63 ± 0.02 ^c^	0.88 ± 0.01 ^a^	34.98 ± 0.82 ^c^
0.8%	2453.67 ± 69.92 ^a^	0.73 ± 0.03 ^b^	0.89 ± 0.03 ^a^	46.79 ± 1.45 ^b^
Natural pork rinds (boiled for 30 min)	2078.67 ± 55.81 ^b,c^	0.93 ± 0.02 ^a^	0.92 ± 0.03 ^a^	52.27 ± 1.64 ^a^

Different lowercase superscript letters (a–e) in the same column denote significant differences (*p* < 0.05). All data are presented as mean values ± standard error (*n* = 3).

**Table 2 molecules-27-03103-t002:** The preliminary sensory score of the plant-based pork rinds containing different SPS and KGM concentrations (v_1_, v_2_, v_3_, and v_4_ represented very good, good, medium, and bad, respectively).

Group	Appearance				Chewiness				Stickiness				Cohesiveness			
SPS	v_1_	v_2_	v_3_	v_4_	v_1_	v_2_	v_3_	v_4_	v_1_	v_2_	v_3_	v_4_	v_1_	v_2_	v_3_	v_4_
10%	0	0	0	10	0	0	1	9	0	0	0	10	0	0	2	8
20%	0	4	4	2	0	2	5	3	0	1	6	3	1	4	5	0
30%	2	6	2	0	3	4	2	0	4	5	1	0	1	7	2	0
40%	2	7	1	0	5	5	0	0	7	3	0	0	1	8	1	0
50%	0	2	5	3	0	5	5	0	9	1	0	0	3	4	3	0
KGM	v_1_	v_2_	v_3_	v_4_	v_1_	v_2_	v_3_	v_4_	v_1_	v_2_	v_3_	v_4_	v_1_	v_2_	v_3_	v_4_
0%	0	1	4	5	1	4	5	0	0	0	5	5	1	5	4	0
0.2%	1	4	5	0	0	3	7	0	1	6	2	1	0	7	3	0
0.4%	4	6	0	0	2	4	4	0	3	7	0	0	1	7	2	0
0.6%	0	6	4	0	1	6	3	0	7	3	0	0	1	8	1	0
0.8%	0	3	6	1	1	5	4	0	9	1	0	0	2	7	1	0

## Data Availability

The data presented in this study are available on request from the corresponding author.

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
