# Peer review of "Mimic Pork Rinds from Plant-Based Gel: The Influence of Sweet Potato Starch and Konjac Glucomannan"

_molecules, 2022, doi:10.3390/molecules27103103_

Round 1
Reviewer 1 Report
on line 363 "The gelling mechanism and interior layout of the plant-based pork rinds are hypothesized: the three-dimensional network structures of samples result from the interaction between the SPS, SPI, and soybean oil with the KGM-SPS complexes filling the pores and the NaHCO3 promoting KGM and SPS interaction between the SPS, SPI, and soybean oil with the 365
tion. " is mentioned. Couldn't the interaction between KGM and SPS be demonstrated with FT-IR.?
The photographic image of the prepared gels can be given in the manuscript.
Are there any cross-linking in the prepared gels? There is a cross-linking between which materials should be clearly shown as a figure.
Reviewer 2 Report
Given in the manuscript; highlighted yellow

Reviewer 3 Report
Manuscript ID: Molecules 2022,27, x. https://doi.org/10.3390/xxxxx
Title: Mimic pork rinds from plant-based gel: the influence of sweet potato starch and konjac glucomannan concentrations
Authors: Qibo Zhang, Lu Huang, He Li, Di Zhao, Jinnuo Cao, Yao Song, and Xinqi Liu
- Overview and general recommendation:
This study looked at how sweet potato starch (SPS) and konjac glucomannan (KGM) affect the textural, color, sensory, rheological, and microstructures of plant-based pork rinds. Plant-based gels were created by combining soy protein isolate (SPI), soy oil, and NaHCO3 with varying concentrations of SPS and KGM. The addition of SPS and KGM had a significant effect on the TPA parameters. However, SPS and KGM concentrations had no significant effect on the observed samples. Moreover, the sensory evaluation indicated that the 40% SPS samples and the 0.4% KGM samples had attributes the most similar to natural pork rinds. The addition of SPS and KGM affected the gel structure making it firmer. This was proved by scanning electron microscopy (SEM) images that revealed the formation of a more compact gel structure. The interaction of SPS, SPI, and soybean oil may have resulted in the three-dimensional network structures of the samples, according to this study, while the addition of KGM and NaHCO3 enabled a more stable gel structure.
The subject of the manuscript is quite interesting, and up to date with research avenues regarding plant-based food production. The studies are comprehensively described and supported by the literature.
Below I give my concerns, that need revision.
- Major comments:
- The abstract could be more specific regarding what methods were used to evaluate the chosen samples attributes. Additionally, it will be better to mention that this study proposed a possible schematic model for the gelling mechanism and internal layout of plant-based pork rinds.
- Line 23 – 24: “The samples displayed similar structures to natural pork rinds, which provided a similar texture and mouthfeel.” – based on the discussion not all the samples had similar attributes. the authors need to specify which samples had the best attributes according to the findings.
- Line 77 – 78: “To the best of our knowledge, there is no report regarding the molecule interaction within plant-based pork rinds.” – since the authors in this study didn’t evaluate the molecular interactions within plant-based pork rinds. I don’t think it’s a good argument to explain the purpose of this specific study.
- Line 168: “Frequency scanning” is it still a frequency scanning or its temperature scanning? Additionally, in my opinion there is no difference between figure 3 and figure 4.
- Line 250 – 251: “glucomannan concentrations >80%” – what about the rest (20%)?
- In my opinion the conclusion is too long it’s more a summary.
- Minor comments:
Figure 6: the attention to details should be better. In my opinion the schematic model at the beginning is confusing, cause the arrow from the emulsion in pointing only towards SPS (amylopectin and amylose). It would be better to showcase that the emulsion contains the other ingredients too.
Author Response
Please see the attachment.

This manuscript is a resubmission of an earlier submission. The following is a list of the peer review reports and author responses from that submission.